# Multi-Modal Inductive Framework for Text-Video Retrieval

Qian Li
Beijing University of Posts
and Telecommunications
Beijing, China
li.qian@bupt.edu.cn

Yucheng Zhou
University of Macau
Macau, China

Cheng Ji
Beihang University
Beijing, China

Feihong Lu
Beihang University
Beijing, China

Jianian Gong
Beihang University
Beijing, China

Shangguang Wang*
Beijing University of Posts
and Telecommunications
Beijing, China

Jianxin Li
Beihang University
Beijing, China

## Abstract

Text-video retrieval (TVR) identifies relevant videos based on textual queries. Existing methods are limited by their ability to understand and connect different modalities, resulting in increased difficulty in retrievals. In this paper, we propose a generation-based TVR paradigm facilitated by LLM distillation to better learn and capture deep retrieval knowledge for text-video retrieval, amidst the rapid evolution of Large Language Models. Specifically, we first design the fine-tuning large vision-language model that leverages the knowledge learned from language models to enhance the alignment of semantic information between the text and video modalities. It also incorporates an inductive reasoning mechanism, which focuses on incorporating important temporal and spatial features into the video embeddings. We further design question prompt clustering to select the most important prompts, considering their contribution to improving retrieval performance. Experimental results show that our approach achieves excellent performance on two benchmark datasets compared to its competitors.

## CCS Concepts

• **Information systems** → **Information retrieval**.

## Keywords

Text-video retrieval, Multi-modal inductive, Fine-tuning LLM.

**ACM Reference Format:**
Qian Li, Yucheng Zhou, Cheng Ji, Feihong Lu, Jianian Gong, Shangguang Wang, and Jianxin Li. 2024. Multi-Modal Inductive Framework for Text-Video Retrieval. In *Proceedings of the 32nd ACM International Conference on Multimedia (MM '24), October 28-November 1, 2024, Melbourne, VIC, Australia.* ACM, New York, NY, USA, 10 pages. https://doi.org/10.1145/3664647.3681024

*Shangguang Wang is the corresponding author.

## 1 Introduction

Text-Video Retrieval (TVR) has emerged as an indispensable task that aims to deliver relevant videos based on their textual queries [51, 58]. It utilizes the fusion of text and video modalities to enhance the accuracy and diversity of search results. By effectively integrating these two modalities, TVR enables users to find videos that match their queries more effectively. It plays a crucial role in a range of applications, including video recommendation systems [16, 39] and video question answering systems [5, 49].

Text-video retrieval is different from uni-modal tasks like conventional ad-hoc retrieval [14, 47], as it involves operating across different modalities. The goal of text-video retrieval is to accurately identify videos that are relevant to given textual queries. However, this task is challenging due to the heterogeneity between text and video modalities. Traditional methods [21, 54] for video retrieval often rely on metadata or manually annotated tags, which have limitations in capturing the rich semantics and nuances present in videos. Some approaches [2, 17] utilize textual information alone, such as keyword matching or semantic techniques. Others leverage visual features extracted from videos to improve retrieval accuracy[26, 45]. However, these methods struggle to handle the vast amount of textual and visual data available, leading to suboptimal retrieval results. Cross-modality semantic representation and alignment are at the core of the text-video retrieval task [25, 52]. Existing work can be mainly divided into two categories: one focuses on cross-modal semantic representation, and the other focuses on cross-modal semantic alignment. Methods like CLIP [40] and CLIP4CLIP [35] embed the textual query and the video into a shared semantic space to calculate similarity, but they fail to capture fine-grained interactions [43]. Other methods employ attention mechanisms to capture the interaction between textual words and video frames, achieving significant performance improvements [1, 36]. LEAN [27] and DGL [53] utilize strategy fostering richer interactions between text and video content. However, there is still a need to explore cross-modality semantic learning in a more systematic manner.

In this paper, amidst the rapid advancement of Large Language Models (LLMs), we break free from conventional thought patterns and introduce a **generation-based TVR paradigm through LLM distillation** (Figure 1), aiming to revisit and reassess the future development and potential value of TVR. Unlike existing methods that rely on pre-defined or context features, our approach designs a Multi-Modal Inductive framework to better capture the semantics of textual queries and video contents for Text-Video Retrieval

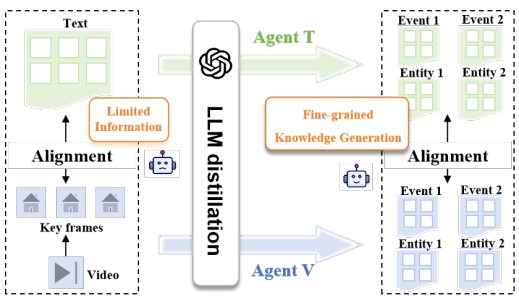

**Figure 1: We break free from conventional thought patterns and introduce a generation-based text-video retrieval paradigm through LLM distillation.**

termed as **MMI-TVR**[1]. Specifically, we first design the fine-tuning large vision-language model to learn the associations between textual queries and video contents, capturing the nuances and context of both modalities. To incorporate important temporal and spatial features into the video embeddings, we employ an inductive reasoning mechanism that utilizes the learned knowledge from the language model. This attention mechanism helps to align the semantic information between the text and video modalities, improving retrieval performance. We further design the fine-grained knowledge generation component to generate prompts that are relevant and informative and facilitate retrieval and problem-solving. These prompts serve the purpose of generating more features specifically tailored for the TVR task. Furthermore, we employ question prompt clustering to select the most important prompts, considering their contribution to improving retrieval performance. Our extensive experiments on benchmark datasets demonstrate that our approach outperforms strong competitors and achieves excellent text-video retrieval performance. Our contributions can be summarized as follows.

- To our best knowledge, we are the first to fine-tune the large vision-language model to understand better cross-modal semantic relationships to enhance the TVR task.
- We design an inductive reasoning mechanism to incorporate important temporal and spatial features and fine-grained knowledge generation to incorporate retrieval reasoning information.
- Experimental results indicate that the framework achieves state-of-the-art performance on the public text-video retrieval datasets.

## 2 Related Works

### 2.1 Text-Video Retrieval

In recent years, significant advancements have been made in the field of Text-Video Retrieval (TVR). Several methods have been proposed to address the challenge of retrieving relevant videos based on textual queries [18, 35]. These methods involve the extraction and encoding of multi-modal features from videos, including visual, and textual information. The integration of multi-modal features and the use of pre-trained models, such as the Multi-modal Transformer (MMT) [12] and CLIP [40], have significantly improved the

performance of TVR systems. The Multi-modal Transformer (MMT) model [12] effectively captures and integrates information from different modalities to enhance retrieval performance. CLIP (Contrastive Language-Image Pretraining) [40] is a pre-trained model that learns joint representations of images and texts. By leveraging the pre-trained CLIP model, researchers have achieved notable improvements in video retrieval performance [11, 35]. The above methods demonstrate the effectiveness of joint learning multi-modal features for the text-video retrieval task. Nevertheless, all of these methods ignore the fine-grained multi-modal information, as well as the object features.

Another effective strategy in text-video retrieval is to focus on fine-grained matching and alignment between video and text [7, 29]. This approach aims to capture detailed correspondences between the two modalities, leading to more accurate retrieval results. The Context-Aware Mixture of Experts (CAMoE) network proposed by Cheng et al. [4] leverages a mixture of expert models to align video features with various textual aspects, enhancing retrieval accuracy. The T2VLAD method [46] employs global-local alignment to better capture spatial and temporal information in videos. The Hierarchical Transformer [31] performs cross-modal hierarchical matching to capture semantic and temporal dependencies between video frames and textual queries. These fine-grained matching and alignment approaches have shown effectiveness in improving retrieval performance [28, 38]. Furthermore, some approaches have also introduced additional meta information, such as video captions [48], video titles [9], and object features [29], to facilitate text-video retrieval. Nevertheless, the above existing methods ignore fine-grained alignment between text and video pairs and in turn may constrain the effectiveness of the text-video retrieval.

Building on these foundations, our paper, set against the backdrop of the rapid advancement of LLMs [55], introduces a generation-based TVR paradigm through LLM distillation, as shown in Figure 1. Our approach aims to revisit and reassess the future development and potential value of TVR, marking a departure from conventional methodologies.

### 2.2 Large Language Models

The development of NLP has boost remarkable progress in Multi-modal Large Language Models (MLLMs) [30, 59]. Owing to the robust background knowledge and inferential capabilities of multi-modal systems, MLLMs are proficient in comprehensively understanding the correspondences between visual and textual inputs. Consequently, this enables them to achieve relatively favorable results in tasks such as text-video retrieval. Some methods use MLLMs to enhance the original data [44, 57], such as generating descriptions of images using MLLMs, thereby augmenting the original data and improving the ability of image-text retrieval models. FuseCap [42] uses LLM to fuse the output of such visual experts with raw captions, producing comprehensive image descriptions that are then used to enhance the model's training data.

## 3 PRELIMINARIES

DEFINITION 1. ***Text-Video Retrieval (TVR).*** *Given a textual query q represented as a set of words, and a collection of videos $V = \{v_1, v_2, \ldots, v_n\}$, where each video $v_i$ consists of a sequence of frames,*

---

[1]The source code is available at https://anonymous.4open.science/r/MMI-TVR-9119.

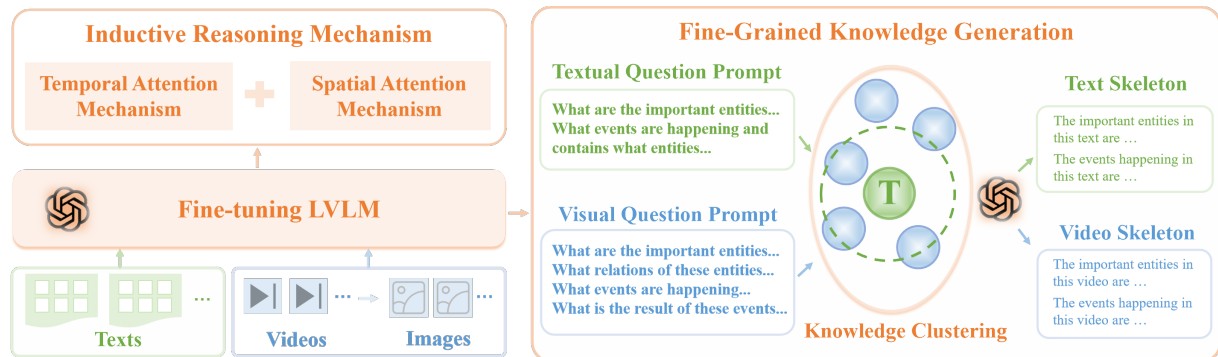

**Figure 2: The MMI-TVR framework for the text-video retrieval. The Fine-tuning large vision-language model is trained to comprehend the associations between the textual and visual modalities. Fine-grained knowledge generation generates prompts that promote thinking and reasoning by clustering question prompts.**

*and a set of corresponding textual descriptions $T = \{t_1, t_2, \ldots, t_n\}$, where each description $t_j$ corresponds to the video $v_j$. The objective of the TVR task is to retrieve a subset of relevant videos $V'$ from $V$, which are considered to have high relevance to the query $q$. Alternatively, the goal is to retrieve a subset of relevant textual descriptions $T'$ from $T$, which are highly relevant to the query $q$.*

The task involves matching a given textual query with relevant videos or textual descriptions. The retrieval objective is to retrieve a subset of videos or textual descriptions from a collection that is highly relevant to the query.

## 4 Methodology

This section introduces our proposed multi-modal inductive text-video retrieval model (MMI-TVR). As shown in Figure 2, MMI-TVR consists of the following two modules: **1) Fine-tuning large vision-language model.** With a large vision-language architecture, the model is trained to comprehend the associations between the textual and visual modalities. The inductive reasoning mechanism contains two attention mechanisms utilized to highlight important temporal and spatial features in video embeddings, which helps to align the semantic information between the text and video modalities, improving retrieval performance. **2) Fine-Grained Knowledge Generation.** It generates prompts that promote structured thinking and reasoning by generating prompts that are relevant and informative and facilitate retrieval. This is accomplished through clustering question prompts and sampling demonstrations that satisfy specific criteria.

### 4.1 Text-Video Large Language Model

To effectively understand and retrieve information from both textual queries and video content, we propose fine-tuning a large visual-language model (LVLM) that leverages transformer-based architecture LLaVA [30]. It has demonstrated remarkable performance in multi-modal NLP tasks, as it excels at capturing long-range dependencies and contextual information. The input for the fine-tuning LLM consists of query text ($T$) and key-images ($I$) extracted from videos ($V$) and videos ($V$). The key-images are selected based on notable disparities [27] in the video content from all frames. To represent the key-images of k-th, we average the weighted frames

between (k-1)-th and k-th in a video as a key-images of k-th input $\mathbf{h}_{I_k} = \sum_{i=N(k-1)}^{N(k)} \alpha_i (\mathbf{W}_I \cdot g(F))$, where $N(k)$ is the number of last frames of key-images of k-th, and the number of key-images represents K, and $F$ is the frames in the video, $\alpha_i$ represents the weight assigned to each frame and caculated by softmax function, and $W_I$ is a trainable weight matrix, and $g(\cdot)$ represents the pre-trained CLIP visual encoder ViT-L/14 [8]. To further represent the video information, we average the weighted sum of all key-images in a video as a video input $\mathbf{h}_v = \sum_{i=1}^{M} \beta_i (\mathbf{W}_V \cdot g(I))$, where $M$ is the total number of key-images in the video, $\beta_i$ represents the weight assigned to each key-image and caculated by softmax function, $W_V$ is a trainable weight matrix. The fine-tuned model is constructed using a transformer-based architecture that utilizes self-attention mechanisms. These mechanisms capture dependencies and relationships between different tokens in input sequences.

*4.1.1 Inductive Reasoning Mechanism.* The inductive attention mechanism serves as a pivotal component in our framework, allowing us to emphasize crucial temporal and spatial features within video embeddings. By computing attention weights between the query representation and the video embeddings, it effectively aligns the semantic information across the text and video modalities.

*Temporal Attention Mechanism.* We further enhance the model's performance by applying a temporal attention mechanism. This mechanism is designed to capture the temporal dynamics of video sequences and align them with the corresponding textual queries. To incorporate temporal attention, we extend the self-attention mechanism of the transformer model to consider the temporal dimension of the video sequences. The temporal attention mechanism can be mathematically described as:

$$\text{TA}(\mathbf{Q}_I, \mathbf{K}_I, \mathbf{V}_I) = \sigma \left( \frac{(\mathbf{Q}_I \cdot \mathbf{W}_{TQ}) \cdot (\mathbf{K}_I \cdot \mathbf{W}_{TK} \cdot \mathbf{H}^T)}{\sqrt{d_k}} \right) \mathbf{V}_I \cdot \mathbf{W}_{TV}, \quad (1)$$

where $\mathbf{Q}_I$, $\mathbf{K}_I$, and $\mathbf{V}_I$ represent the query, key, and value matrices of key-images respectively, and $d_k$ is the dimension of the key vectors. $\mathbf{W}_{TQ}$, $\mathbf{W}_{TK}$, and $\mathbf{W}_{TV}$ are learned weights corresponding to the query, key, and value matrices. $\sigma$ is the softmax function. The temporal matrix $\mathbf{H} \in \mathbb{R}^{K \times K}$ captures the temporal relationships between key-images embeddings and obtains temporal embedding $E_t$,

which allows the model to attend to relevant temporal features. $\mathbf{H}$ is initialized by the combination metric of the positional information of key-images and semantic similarities between the key-images to learn each embedding considering the temporal connection.

*Spatial Attention Mechanism.* In addition to the temporal attention mechanism, we also introduce a spatial attention mechanism to capture the spatial relations within video frames. This mechanism aims to focus on specific regions or objects in the video frames that are relevant to the textual queries. To incorporate spatial attention, we extend the self-attention mechanism to consider the spatial dimension of video frames. The spatial attention mechanism can be mathematically described as:

$$\text{SA}(\mathbf{Q}_I, \mathbf{K}_I, \mathbf{V}_I) = \sigma \left( \frac{(\mathbf{Q}_I \cdot \mathbf{W}_{\text{SQ}}) \cdot (\mathbf{K}_I \cdot \mathbf{W}_{\text{SK}} \cdot \mathbf{G}^T)}{\sqrt{d_k}} \right) \cdot \mathbf{V}_I \cdot \mathbf{W}_{\text{SV}}, \quad (2)$$

where $\mathbf{W}_{SQ}$, $\mathbf{W}_{SK}$, and $\mathbf{W}_{SV}$ are learned weights corresponding to the query, key, and value matrices. The spatial matrix $\mathbf{G} \in \mathbb{R}^{K \times K}$ captures the spatial relationships between regions or objects in the video frames and obtains spatial embedding $\mathbf{h}_s$, allowing the model to attend to specific spatial features. $\mathbf{G}$ is initialized using a strategy that integrates both geometric proximity and semantic similarity among regions or objects within the frames, to ensure that each embedding takes into account the spatial connection. With the incorporation of this attention mechanism, the model effectively emphasizes important temporal and spatial features within the video embeddings. This process significantly improves the alignment of semantic information between the text and video modalities, leading to enhanced retrieval performance and a more robust mapping of textual queries to relevant video content. The final visual embedding $\mathbf{h}_v$ is the fusion of temporal embedding $\mathbf{h}_t$ and spatial embedding $\mathbf{h}_s$ after the two attentions by using the nonlinear transformation.

To predict the masked tokens based on the context provided by the remaining tokens, we mask certain tokens in textual queries and video sequences, and train the model. It enables the model to learn the semantic relationships between textual and visual modalities. The objective function for fine-tuning involves minimizing the negative log-likelihood of predicting the masked video embeddings given the textual query, and vice versa. The objective function for fine-tuning can be represented as:

$$\mathcal{L}_{\text{m}} = -\log P(V_1, \ldots, V_n | T) - \log P(T | V_1, \ldots, V_n), \quad (3)$$

where $\{V_1, \ldots, V_n\}$ represents a sequence of video embeddings, and $T$ represents the corresponding textual query.

## 4.2 Fine-Grained Knowledge Generation

Fine-grained knowledge generation generates knowledge that facilitates reasoning and problem-solving, referring to the process of generating knowledge that can elicit specific information or responses. It encourages coherent and structured thinking, which consists of two main modules: Knowledge generation and Knowledge clustering.

*4.2.1 Knowledge Generation.* Knowledge generation involves generating knowledge in both textual and video formats to facilitate coherent and structured thinking. It generates diverse and targeted

knowledge in both textual and video formats. These prompts serve as the initial stimuli for the subsequent stages of clustering, facilitating coherent reasoning and problem-solving.

*Textual Knowledge Generation.* It is achieved by defining appropriate instruction templates for input into the LLM. These prompts are designed to elicit implicit information by answering specific questions.

```
1. What key concepts are mentioned in this text?
2. Which events are occurring, and which entities are
   involved??
```

This prompt aims to guide the model to identify and extract important entities or words from the given text to gain a better understanding. Multiple questions are generated for each text, capturing different aspects of implicit information relevant to text retrieval and comprehension.

*Visual Knowledge Generation.* The visual knowledge generation focuses on generating prompts that are specific to visual content. Instead of text-based prompts, the algorithm generates prompts tailored to the visual information in a video. A corresponding "Instruction Pool" has been defined, with a few selected examples shown below

```
1. What are the important entities in this video?
2. What are the relationships of these entities in this
   video?
```

Other settings for visual knowledge generation are consistent with those of textual knowledge generation.

*4.2.2 Knowledge Clustering.* The knowledge clustering involves partitioning a set of question prompts into a small number of clusters. This partitioning is based on the number of demonstrations that question prompts can support. A diversity-based clustering approach is used, ensuring the mitigation of errors and the coverage of different types of prompts.

To cluster questions, a diversity-based approach is applied. Instead of clustering entire demonstrations (question, answer and retrieval result pairs), we only cluster the questions prompts themselves. We calculate a vector representation for each question by the fine-tuning large vision-language model in Section 4.1. These question representations are then processed using k-means clustering. For each cluster $i$, we sort the questions into a list $\mathbf{q}^i = \left[ q_1^i, q_2^i, \ldots \right]$ in ascending order of distance to the cluster center. When generating demonstrations, we give preference to the most typical question in each cluster. The objective function for clustering the question prompts is as follows:

$$\text{minimize} \sum_{i=1}^{k} \sum_{q_j^i \in \mathbf{q}^i} \text{d}(q_j^i, c_i), \quad (4)$$

$$\text{d}(q^i, c_i) = \left\| \frac{1}{|q^i|} \sum_{w \in q_i} \mathbf{h}(w) - \mathbf{h}(c_i) \right\|, \quad (5)$$

where $k$ represents the number of clusters, $\|\|$ means Euclidean norm, $\mathbf{q}^i$ is the list of questions in cluster $i$, and $c_i$ is the centroid (or center) of cluster $i$. $\mathbf{h}(w)$ and $\mathbf{h}(c_i)$ represents the embedding of word $w$ and center $c$. The goal is to minimize the distance between

each question $q_j^i$ and its cluster center $c_i$. The distance function $d(q^i, c_i)$ measures the similarity between a question $q^i$ and the cluster center $c$. It is computed as the Euclidean distance between the average embeddings of the words in the question $q^i$ and the cluster center $c^i$.

To select a representative question prompt from each cluster and generate a reasoning chain as the skeleton using the model, we use demonstration sampling. Formally, for each cluster $i$ ($i = 1, \ldots, k$), a demonstration $d^i$, consisting of a question, answer, and retrieval result, is generated. Starting with the closest question to the cluster center, a prompted input is formulated as Section 4.2.1. The input is then fed into the fine-tuning model to produce a reasoning chain comprising an answer $a_j^i$ and a retrieval result $r_j^i$. A candidate demonstration $d_j^i$ for cluster $i$ is as:

$$d_j^i = [Q : q_j^i, \ A : a_j^i \circ r_j^i], \tag{6}$$

where $d_j^i$ is constructed by concatenating the question $q_j^i$, answer $r_j^i$, and retrieval result $a_j^i$. The algorithm generates prompts that encourage structured thinking by clustering prompts and sampling demonstrations that satisfy specific criteria.

The objective function for demonstration sampling is as:

$$\mathrm{argmin}_{d_j^i} \left( \mathrm{len}(q_j^i), \mathrm{len}(a_j^i), \mathrm{count}(\nabla, a_j^i) \right). \tag{7}$$

The objective for demonstration sampling is to minimize the length of the question, the length of the answer, and the count of the answer terms. $\nabla$ is a predefined set of answer terms. Through fine-grained knowledge generation, the model generates two types of skeletons: the Text Skeleton and the Video Skeleton.

*Text Skeleton.* The text skeleton is the generation of a text skeleton, which involves extracting key semantic concepts and coherently structuring them. It also contains the answer to the textual question prompt.

*Video Skeleton.* It includes the generation of a video skeleton, which aims to distill the salient visual and temporal features of the video data. This process identifies important objects, actions, and their temporal boundaries within the video by the answer to the visual question prompt.

## 4.3  Joint Objective

The objective is to bridge the gap between text and video modalities and facilitate the retrieval of relevant videos based on textual queries. To achieve this objective, we employ the cosine similarity metric, to compute a similarity score between the text representation ($\mathbf{h}_T$) and the video embeddings ($\mathbf{h}_V$) obtained from the videos in the dataset. The text-video retrieval process can be formalized as follows:

$$\mathcal{L}_r = sim(\mathbf{h}_T, \mathbf{h}_v). \tag{8}$$

In order to further improve the performance of fine-tuned large-scale language-visual models for text video retrieval tasks, we adopt the LVLM encoder $Enc()$ to evaluate the similarity between the generated Text Skeleton ($T_s$) and Video Skeleton ($V_s$).

$$\mathcal{L}_s = Enc(P, T_s, V_s), \tag{9}$$

where $P$ is the prompt for evaluate the similarity. The prompt is that *"Given a probability value of [0,1], evaluate the probability that the Textual Skeleton and Video Skeleton are describing a scene."* In this way, the model can understand and mine the fine-grained semantic connections between text queries and video content, thereby providing richer and more accurate semantic information for retrieval.

The overall joint objective is formed by combining the two losses in the following formulation:

$$\mathcal{L} = \lambda_r \mathcal{L}_r + \lambda_s \mathcal{L}_s, \tag{10}$$

where $\lambda_r$, and $\lambda_s$ are trade-off parameters. The optimization is performed using a mini-batch strategy.

## 5  Experiments

### 5.1  Dataset and Evaluation Metric

We conducted experiments on five widely used Text-Video Retrieval datasets: MSR-VTT [50] is a well-known dataset specifically designed for open-domain video captioning. MSVD [3] is another widely used dataset for text-video retrieval tasks. LSMDC [41] is a dataset created in a joint effort by Johns Hopkins University and FAIR. DiDeMo [20] consists of 10K Flickr videos annotated with 40K text captions. ActivityNet [19] contains 20K YouTube videos annotated with 100K sentences, with 10K videos in the training set.

To evaluate the performance of our models, we reported several official evaluation metrics that are widely used in the retrieval literature, including R@1, R@5, R@10, MdR, and MnR. These metrics provide a comprehensive assessment of the models' performance in terms of retrieval accuracy and ranking evaluation. They have been extensively employed in previous works [15, 36].

### 5.2  Comparision Methods

We compare our method with three traditional TVR models **CE** [32], **SSB** [37], and **FROZEN** [1], which focus on multi-modal modeling, and five CLIP-based TVR models **CLIP4Clip** [35], **CLIP2Video** [11], **X-CLIP** [36], **ClipBERT** [24], and **CenterCLIP** [56], which focus on intergrate more multi-modal information, and five cross-modality learning TVR models **MIL-NCE** [33], **DiCoSA** [23], **LEAN** [27], **TS2-Net** [34], **EMCL** [22], **X-Pool** [13], **TT-CE** [6], **DGL** [53], and **UATVR** [10], which focus on cross-modality semantic representation and alignment among the videos and the texts.

### 5.3  Implementation Details

We used PyTorch[2] as a deep learning framework to develop the TVR. All experiments were conducted on a server with four GPU (Tesla V100). The LLava version is llava-v1.5-13b in huggingface[3] for text and video initialization and the dimension was set to 768. Training is performed using Adam optimizer with a learning rate of 0.001, and learning rate is 3e-5. We use a batch size of 512 and apply a dropout rate of 0.1 to prevent overfitting. The text and visual instruction pool have 50 questions respectively. The number of clusters k is calculated by the knowledge clustering. For hyperparameters, the best coefficients $\lambda_r$, $\lambda_s$ are 0.6, and 0.3.

---

[2] https://pytorch.org/
[3] https://huggingface.co/liuhaotian/llava-v1.5-13b

**Table 1: Main experiments. The best and second-best results are highlighted in bold and underlined, "–" means results are not available. '↑' denotes that higher is better. "↓" denotes that lower is better.**

| Dataset | Method | Text-to-Video Retrieval | | | | | Video-to-Text Retrieval | | | | |
|---|---|---|---|---|---|---|---|---|---|---|---|
| | | R@1↑ | R@5↑ | R@10↑ | MdR↓ | MnR↓ | R@1↑ | R@5↑ | R@10↑ | MdR↓ | MnR↓ |
| MSR-VTT | CE [32] | 20.9 | 48.8 | 62.4 | 6.0 | 28.2 | 20.6 | 50.3 | 64.0 | 5.3 | 25.1 |
| | SSB [37] | 27.4 | 56.3 | 67.7 | 3.0 | - | 26.6 | 55.1 | 67.5 | 3.0 | - |
| | FROZEN [1] | 31.0 | 59.5 | 70.5 | 3.0 | - | - | - | - | - | - |
| | CLIP4Clip-meanP [35] | 43.1 | 70.4 | 80.8 | 2.0 | 16.2 | 43.1 | 70.5 | 81.2 | 2.0 | 12.4 |
| | CLIP2Video [11] | 45.6 | 72.6 | 81.7 | 2.0 | 14.6 | 43.5 | 72.3 | 82.1 | 2.0 | 10.2 |
| | X-CLIP (ViT-B/16) [36] | 49.3 | 75.8 | 84.8 | 2.0 | 12.2 | 48.9 | 76.8 | 84.5 | 2.0 | 8.1 |
| | MIL-NCE [33] | 47.2 | 73.0 | 82.8 | 2.0 | 13.9 | 46.3 | 74.1 | 84.8 | 2.0 | 8.8 |
| | DiCoSA [23] | 47.5 | 74.7 | 83.8 | 2.0 | 13.2 | 46.7 | 75.2 | 84.3 | 2.0 | 8.9 |
| | DGL [53] | 48.3 | 71.8 | 80.6 | - | 13.4 | 45.7 | 74.0 | 82.9 | - | 10.9 |
| | LEAN [27] | 50.6 | 77.1 | 85.2 | 2.0 | 11.4 | 49.1 | 78.2 | 86.7 | 2.0 | 7.3 |
| | **MMI-TVR (Ours)** | **52.4** | **78.0** | **87.6** | 2.0 | **11.1** | **51.2** | **79.5** | **87.4** | 2.0 | **6.8** |
| MSVD | CE [32] | 19.8 | 49.0 | 63.8 | 6.0 | 23.1 | - | - | - | - | - |
| | SSB [37] | 28.4 | 60.0 | 72.9 | 4.0 | - | - | - | - | - | - |
| | FROZEN [1] | 33.7 | 64.7 | 76.3 | 3.0 | - | - | - | - | - | - |
| | CLIP4Clip-meanP [35] | 46.2 | 76.1 | 84.6 | 2.0 | 10.0 | 56.6 | 79.7 | 84.3 | 1.0 | 7.6 |
| | CLIP2Video [11] | 47.0 | 76.8 | 85.9 | 2.0 | 9.6 | 58.7 | 85.6 | 91.6 | 1.0 | 4.3 |
| | X-CLIP (ViT-B16) [36] | 50.4 | 80.6 | - | - | 8.4 | 66.8 | 90.4 | - | - | 4.2 |
| | MIL-NCE [33] | 47.5 | 78.0 | 86.6 | 2.0 | 9.3 | 70.2 | 88.1 | 92.7 | 1.0 | 6.0 |
| | DiCoSA [23] | 47.4 | 76.8 | 86.0 | 2.0 | 9.1 | - | - | - | - | - |
| | UATVR(ViT-B16) [10] | 49.7 | 79.0 | 87.3 | 2.0 | 8.9 | - | - | - | - | - |
| | LEAN [27] | 52.1 | 81.9 | 87.0 | 2.0 | 7.4 | 71.3 | 91.7 | 93.8 | 1.0 | 4.1 |
| | **MMI-TVR (Ours)** | **53.8** | **82.5** | **87.7** | 2.0 | **7.1** | **73.0** | **93.4** | **94.7** | 1.0 | **3.6** |
| LSMDC | TS2-Net [34] | 23.4 | 42.3 | 50.9 | 9.0 | 56.9 | - | - | - | - | - |
| | FROZEN [1] | 15.0 | 30.8 | 39.8 | 20.0 | - | - | - | - | - | - |
| | CLIP4Clip (seqLSTM) [35] | 21.6 | 41.8 | 49.8 | - | 58.0 | 20.9 | 40.7 | 49.1 | - | 53.9 |
| | EMCL [22] | 23.9 | 42.4 | 50.9 | 10.0 | - | 22.2 | 40.6 | 49.2 | - | - |
| | X-CLIP [36] | 23.3 | 43.0 | - | - | 56.0 | 22.5 | 42.2 | - | - | 50.7 |
| | CenterCLIP [56] | 21.9 | 41.1 | 50.7 | - | 55.6 | 21.1 | 41.2 | 50.2 | - | 51.0 |
| | X-Pool [13] | 25.2 | 43.7 | 53.5 | 8.0 | 53.2 | 22.7 | 42.6 | 51.2 | - | 47.4 |
| | **MMI-TVR (Ours)** | **26.6** | **44.6** | **54.1** | 7.0 | **51.8** | **24.3** | **43.2** | **53.9** | 10.0 | **45.3** |

## 5.4 Main Results

To validate the effectiveness of our proposed model, we present comprehensive results in Table 1, which reflect the overall performance of our model across various evaluation metrics. From the table, we can observe that: 1) Our model consistently outperforms all the baseline models across all evaluation metrics. In particular, we achieve a notable improvement of at least 1.7% on R@1 for TVR and 0.7% on R@10. This clearly indicates that our model is capable of effectively fine-tuning large language models for each text query and video, enabling it to capture intricate correlations between different modalities with great accuracy. 2) When compared to fine-grained matching and alignment TVR models, our approach exhibits superior performance. By effectively capturing the underlying associations between textual and visual modalities, our model surpasses the performance of these models, showcasing

the effectiveness of our proposed methodology. 3) Our model also outperforms existing multi-modal transformer-based TVR models. By leveraging the benefits of both TVR large vision-language model and fine-grained knowledge generation, our approach demonstrates improved performance compared to these models. This suggests the advantages of our model in handling complex relationships between different modalities. 4) It is worth noting that our model even surpasses graph neural networks designed specifically for TVR tasks. This further supports the fact that our fine-tuning large language model can capture intricate correlations and generalize effectively, leading to superior performance. 5) By conducting experiments on multiple text-video retrieval datasets, our model consistently performs the best. This further reinforces the effectiveness and robustness of our model across different scenarios and datasets. All these observations highlight the efficacy of our model in leveraging multi-modal information and capturing complex correlations.

**Table 2: Main experiments of text-to-video retrieval on the ActivityNet and DiDeMo datasets.**

| Method | Text-to-video retrieval on ActivityNet | | | | | Text-to-video retrieval on DiDeMo | | | | |
|---|---|---|---|---|---|---|---|---|---|---|
| | R@1↑ | R@5↑ | R@10↑ | MdR↓ | MnR↓ | R@1↑ | R@5↑ | R@10↑ | MdR↓ | MnR↓ |
| CE [32] | 18.2 | 47.7 | 63.9 | 6.0 | 23.1 | 16.1 | 41.1 | - | 8.3 | 43.7 |
| ClipBERT [24] | 21.3 | 49.0 | 63.5 | 6.0 | - | 24.0 | 48.0 | 60.8 | 6.0 | - |
| TT-CE [6] | 23.5 | 57.2 | - | 4.0 | - | 21.6 | 48.6 | 62.9 | 6.0 | - |
| CLIP4Clip [35] | 40.5 | 72.4 | 83.6 | 2.0 | 7.5 | 42.8 | 68.5 | 79.2 | 2.0 | 18.9 |
| TS2-Net [34] | 41.0 | 73.6 | 84.5 | 2.0 | 8.4 | 41.8 | 71.6 | 82.0 | 2.0 | 14.8 |
| UATVR [10] | - | - | - | - | - | 45.8 | 73.7 | 83.3 | 2.0 | 13.5 |
| DiCoSA [23] | 42.1 | 73.6 | 84.6 | 2.0 | 6.8 | 45.7 | 74.6 | 83.5 | 2.0 | 11.7 |
| **MMI-TVR (Ours)** | **45.3** | **75.1** | **87.9** | **2.0** | **5.7** | **51.2** | **76.0** | **86.2** | **2.0** | **10.5** |

**Table 3: Variant experiments on the MSR-VTT and MSVD datasets. "w/o" means removing corresponding module. "repl." means replacing corresponding module with the other. '↑' denotes that higher is better. "↓" denotes that lower is better.**

| Variants | Text-to-Video Retrieval | | | | | Video-to-Text Retrieval | | | | |
|---|---|---|---|---|---|---|---|---|---|---|
| | R@1↑ | R@5↑ | R@10↑ | MdR↓ | MnR↓ | R@1↑ | R@5↑ | R@10↑ | MdR↓ | MnR↓ |
| **MMI-TVR (Ours) on MSR-VTT** | **52.4** | **78.0** | **87.6** | **2.0** | **11.1** | **51.2** | **79.5** | **87.4** | **2.0** | **6.8** |
| w/o Fine-tuning LVLM | 51.3 | 77.6 | 86.7 | 4.0 | 11.4 | 49.1 | 77.3 | 85.2 | 3.0 | 6.9 |
| w/o Inductive Reasoning Mechanism | 51.8 | 77.1 | 86.9 | 4.0 | 11.7 | 50.2 | 78.0 | 86.3 | 3.0 | 7.3 |
| w/o Temporal Attention Mechanism | 52.1 | 77.8 | 87.3 | 3.0 | 11.6 | 50.3 | 78.4 | 86.9 | 3.0 | 7.0 |
| w/o Spatial Attention Mechanism | 52.4 | 77.2 | 87.1 | 3.0 | 11.5 | 50.7 | 78.6 | 86.8 | 3.0 | 7.1 |
| repl. Self Mechanism | 52.1 | 77.0 | 87.2 | 3.0 | 11.9 | 50.3 | 78.4 | 86.2 | 3.0 | 7.6 |
| w/o Fine-Grained Knowledge Generation | 51.5 | 77.0 | 86.2 | 4.0 | 11.8 | 49.2 | 77.5 | 85.7 | 4.0 | 7.2 |
| w/o Textual Knowledge Generation | 52.3 | 77.2 | 87.0 | 3.0 | 11.7 | 50.4 | 78.6 | 86.9 | 3.0 | 7.3 |
| w/o Visual Knowledge Generation | 52.2 | 77.6 | 87.5 | 3.0 | 11.3 | 50.6 | 78.5 | 86.2 | 3.0 | 7.6 |
| w/o Knowledge Clustering | 52.7 | 77.3 | 87.4 | 3.0 | 11.5 | 50.8 | 78.0 | 86.5 | 3.0 | 7.5 |
| **MMI-TVR (Ours) on MSVD** | **53.8** | **82.5** | **87.7** | **2.0** | **7.1** | **73.0** | **93.4** | **94.7** | **1.0** | **3.6** |
| w/o Fine-tuning LVLM | 52.7 | 80.2 | 85.0 | 4.0 | 5.1 | 71.3 | 91.5 | 92.6 | 3.0 | 2.4 |
| w/o Inductive Reasoning Mechanism | 53.4 | 81.1 | 86.5 | 3.0 | 6.8 | 72.0 | 92.3 | 93.9 | 2.0 | 2.7 |
| w/o Temporal Attention Mechanism | 53.8 | 81.0 | 86.6 | 3.0 | 6.7 | 72.2 | 92.3 | 93.8 | 3.0 | 2.1 |
| w/o Spatial Attention Mechanism | 53.0 | 81.8 | 86.4 | 2.0 | 6.2 | 72.1 | 92.6 | 93.9 | 2.0 | 2.5 |
| repl. Self Mechanism | 53.0 | 81.8 | 86.4 | 2.0 | 6.2 | 72.1 | 92.6 | 93.9 | 2.0 | 2.5 |
| w/o Fine-Grained Knowledge Generation | 52.9 | 80.5 | 85.7 | 3.0 | 5.8 | 71.5 | 91.3 | 92.6 | 3.0 | 2.2 |
| w/o Textual Knowledge Generation | 53.7 | 81.4 | 86.7 | 3.0 | 6.9 | 72.2 | 92.1 | 93.3 | 2.0 | 2.8 |
| w/o Visual Knowledge Generation | 53.3 | 81.8 | 86.6 | 2.0 | 6.7 | 72.0 | 92.2 | 93.5 | 3.0 | 2.1 |
| w/o Knowledge Clustering | 53.8 | 81.7 | 86.3 | 3.0 | 6.6 | 72.5 | 92.4 | 93.2 | 2.0 | 2.9 |

## 5.5 Discussion for Model Variants

To investigate the effectiveness of each module in our proposed model, we conducted variant experiments and showcased the results in Table 3.From the table, we can observe that: 1) The impact of the fine-tuning large vision-language model tends to be more significant than that of other modules. This highlights the crucial role of fine-tuned text-video retrieval large language model in effectively understanding and retrieving information from both textual queries and video contents. By leveraging this module, the model gathers more clues and insights. 2) When the Inductive Reasoning Mechanism is removed, a decrease in performance is observed. This finding emphasizes the pivotal role played by this mechanism

in deducing relevant patterns and insights from both textual and visual modalities. 3) When the temporal attention mechanism is replaced, a decline in performance is observed. This indicates that our full model, which incorporates the temporal attention mechanism, is better equipped to capture the complex correlations between textual and visual modalities. 4) The removal of any one of the prompts used in our model leads to a decrease in performance. This demonstrates the usefulness of each prompt by capturing different retrieval semantics and providing valuable information. 5) The removal of the knowledge clustering also results in a decrease in performance. This suggests that the selected prompts are indeed helpful for the text-video retrieval task.

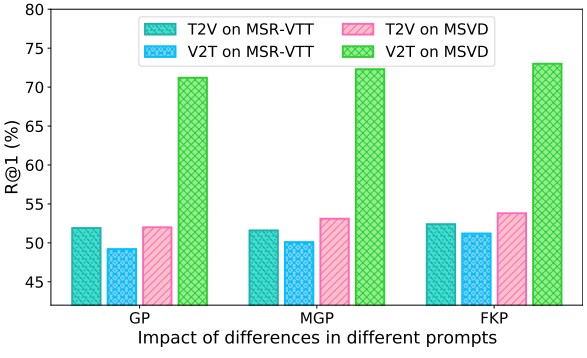

(a) Impact of different prompts on MSR-VTT and MSVD.

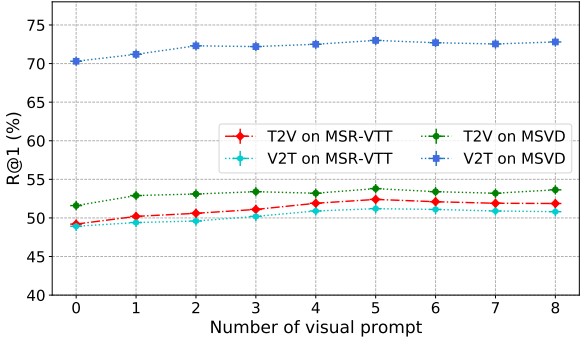

(b) Impact of number in the visual prompts.

**Figure 3: Impact of different knowledge generation and number of prompts.**

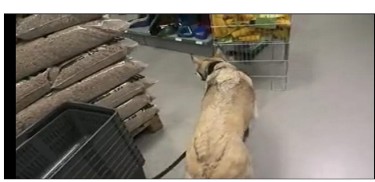

**Key-image of the Video**
**Text: there is a dog is walking on the floor**

**Textual Prompt:**
1. What key concepts are mentioned in this text?
2. Which events are occurring, and which entities are involved?

**Textual Skeleton:**
1. The key concepts include a **dog**, **walking**, and the floor.
2. The primary event is the **dog walking**. Entities involved include **dog** and floor.

**Video Prompt:**
1. What are the important entities in this video?
2. What the entities in the video do?

**Video Skeleton:**
1. Important entities in this video would include the **dog**, the leash, the store aisle, and the items on the shelves.
2. The **dog is walking**, possibly following its owner or exploring the store.

**Figure 4: An example of text-videos pairs for the prompts and the corresponding skeletons.**

## 5.6 Discussion for Knowledge Generation

To analyze the impact of different question prompts on the model's performance, we conducted a thorough comparison by selecting various prompts as input. Specifically, we compared the performance of the model when using the generated prompt by the large language model (GP), fine-grained knowledge prompt (FKP), and the generated prompt by the multi-modal large language model (MGP), as shown in Figure 3 (a). From the figure, we can observe that: 1) The model achieved the highest performance when the fine-grained knowledge prompt module is used as the input prompt. It indicates that the information provided by the FKP module is particularly valuable. The module's ability to generate coherent and contextually relevant prompts seems to have a positive impact on the model's effectiveness in retrieving accurate information. 2) The performance of the model when using the generated prompt by the large language model is comparable to that achieved when using the prompt. This suggests that the prompt generation serves as crucial information for enhancing retrieval.

## 5.7 Discussion for Visual Question Prompts

We further investigated the performance of the model by examining its performance with different numbers of question prompts, as shown in Figure 3 (b).From the figure, we can observe that: 1) We noticed that the performance of the model improves as the number of prompts increases. This suggests that having more effective key prompts is beneficial, as it allows the model to capture more correlated information between the text and video. 2) We found that

the model's performance stabilizes when the number of prompts reaches five. It indicates that within a certain range of prompt numbers, the model is already capable of effectively capturing the correlated information between the text and video.

## 5.8 Discussion for Interpretability

We delve into the interpretability of the multi-modal inductive framework for the text-video retrieval task by examining the outcomes generated from prompts in both text prompt and skeleton pairs, and video prompt and skeleton pairs. An example is shown in Figure 4. From the figure, we can observe that the interpretability aspect is crucial for understanding how the proposed framework processes and aligns different modalities—textual queries with video content—through the responses generated from prompts.

## 6 Conclusion

This paper proposes the multi-modal inductive large vision-language framework for text-video retrieval. It leverages LLM to capture the semantics of textual queries and video contents, enabling accurate and efficient retrieval of relevant videos. We also incorporate an inductive reasoning mechanism to enhance retrieval performance, which focuses on incorporating important temporal and spatial features into the video embeddings. We further design the fine-grained knowledge generation and knowledge clustering to generate retrieval-relevant prompts for generating more features specifically tailored. Our proposed model achieves state-of-the-art performance on the TVR task, outperforming existing methods.

# Acknowledgments

We thank the anonymous reviewers for their insightful comments and suggestions. The corresponding author is Shangguang Wang. The authors of this paper were supported by the NSFC through grant No.62032003 and 62225202.

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
