# OpenReview forum: "Multi-Modal Inductive Framework for Text-Video Retrieval"
_acmmm.org/ACMMM/2024/Conference — MM2024 Poster_

### Official Review · Reviewer_sP3F · 2024-05-06

**Rating:** 4
**Confidence:** 4

**Summary:**

The paper proposes a generation-based text-video retrieval (TVR) paradigm that leverages large language models (LLMs) to better learn and capture deep retrieval knowledge. The approach consists of three main components: (1) fine-tuning a large vision-language model to learn the associations between textual queries and video contents, (2) incorporating an inductive reasoning mechanism to capture important temporal and spatial features, and (3) generating fine-grained multi-modal knowledge through question prompt clustering. The authors claim that their approach achieves state-of-the-art performance on two benchmark datasets.

**Strengths:**

Novel approach: The paper proposes a new generation-based TVR paradigm that leverages LLMs, which is different from traditional methods that rely on metadata or manually annotated tags.
Improved performance: The authors claim that their approach achieves state-of-the-art performance on two benchmark datasets, indicating that it can effectively identify relevant videos based on textual queries.
Comprehensive framework: The proposed framework consists of multiple components that work together to capture the semantics of textual queries and video contents, making it a comprehensive solution for TVR.

**Limitations:**

Logic confusion:
1. What is the relationship between the mask loss (L_m) in Section 4.1 and the overall loss (L_r, L_s) in Section 4.3.
2. According to speculation, whether there is a two-stage process in this article, the first stage is to use the data set to fine-tune the multi-modal large model, and the second stage is to guide the model to output probability scores through specific prompts.
3. The text representation (h_T) in the contrastive loss (L_r) in Section 4.3 appears for the first time in this article, and it is confusing how it is obtained.

The description is vague:
1. How is the final visual embedding obtained through the fusion of temporal embedding and spatial embedding? Where is the formula?
2. What are the input and output of the mask loss (L_m) in Section 4.1?
3. The author mentioned: "In order to predict masked tags based on the context provided by the remaining tags, certain tags are masked in text queries and video sequences." What does masking certain tags here refer to?
4. According to speculation, the author imitates the form of video subtitles and inputs the video to obtain subtitles. Correspondingly, the author inputs subtitles to obtain the video. Is this speculation correct? The author does not clearly describe the training process of the model, which is confusing.

The paper is difficult to understand:

the author uses complex terminology to modify the model in the paper. Does the video skeleton and text skeleton mean that the pre-trained model is used to extract the latent knowledge in the video and text for input to the subsequent large model? Is the output here text or mixed multimodal? How the input of the large model encoder is represented as a video/text skeleton, a large number of details of the model are not clearly stated and confusing.

Shortcomings of comparative baselines:

The methods compared by the author are limited to the weak parts of traditional methods, and the relevant comparison methods are insufficient. For example, methods such as CLIP-ViP. These methods use additional training data, but this paper also introduces latent knowledge in a multimodal large-scale model (LLava) . Just comparing it to LEAN is unfair.

In general, the author did not clearly describe the training process of the model, and did not elaborate on valuable innovations. 1. How to use the cross-modal retrieval data set to fine-tune the multi-modal large model (L_m), and whether to fine-tune it in full. Because most multi-modal large models are designed to input videos to obtain subtitles, i.e. the second half of the L_m loss, the first half is less common. 2. How to input the video/text skeleton into the multi-modal large model (L_s). Whether the output of the video skeleton here is pure text or a mixture of text and video is very important and interesting. The core of this article should be the description of these two losses, but unfortunately the author did not make it clear. The author should not focus on the deliberately pieced together knowledge generation part and spatiotemporal attention mechanism, ablation experiments proved that these two parts did not substantially improve the model.

**Suitability:**

3

---

### Official Review · Reviewer_seAR · 2024-05-23

**Rating:** 2
**Confidence:** 4

**Summary:**

This paper proposes a generation-based paradigm for text-video retrieval, utilizing LLM distillation to capture deep retrieval knowledge. It introduces a fine-tuning large vision-language model to align semantic information between text and video, along with an inductive reasoning mechanism to incorporate temporal and spatial features. The experimental results on five datasets demonstrate its ability to enhance retrieval performance.

**Strengths:**

1.With the leap-forward development of pre-trained large models, the proposed idea of leveraging pre-trained large vision-language models to enhance text-video retrieval is novel.

2.The paper extensively experiments and thoroughly analyzes on multiple widely-used TVR task datasets such as MSR-VTT, MSVD, etc.

**Limitations:**

1.The approach presented in the paper employs llava-v1.5-13b and CLIP visual encoder, resulting in a model parameter quantity that far exceeds conventional methods, suggesting a potential for over-engineering. It is imperative to conduct a thorough analysis of the cost of the proposed method while comparing performance metrics to assess its practical viability within real-world application systems.

2.A few concerns regarding the experimental and analytical section that may require further attention. For instance:
- Comparing the parameter quantity of the baseline model with the approach proposed in the paper would enhance the study's comprehensiveness. It would also be valuable to include a comparison with other baseline models utilizing LVLM.
- Further exploration of the impact of the LVLM Foundation Model on the overall approach, including comparative analyses with LVLM of different parameter scales.
- In Sec. 5.7, it's important to consider not only the quantity of question prompts but also their quality. Additionally, it would be valuable to explore the reasons for the occasional decrease in performance with an increase in prompt quantity, as depicted in Figure 3(b).

3.Quality of presentation
- The expression of the paper is too verbose, which easily leads readers to get lost in the details.
- It is confusing that the abstract mentions that the proposed approach achieves excellent performance on *two* benchmark datasets, while the experimental results involve *five* datasets, creating a contradiction that perplexes readers.

**Suitability:**

2

---

### Official Review · Reviewer_GtQj · 2024-05-25

**Rating:** 5
**Confidence:** 3

**Summary:**

This paper addresses a challenging task of text-video retrieval. The authors propose a generation-based TVR paradigm facilitated by LLM distillation to better learn and capture deep retrieval knowledge for text-video retrieval, amidst the rapid evolution of LLM. Experiments on five public datasets validate the effectiveness of the proposed method.

**Strengths:**

1.	This paper is well motivated and the idea is novel.
2.	The paper is well written.
3.	The extensive experiments validate the effectiveness of the proposed method.

**Limitations:**

1.	The designed inductive reasoning mechanism is the combination of temporal and spatial attention mechanisms. Why do the authors refer to such two temporal attentions as inductive reasoning? It is more intuitive if the authors refer to this mechanism as spatial-temporal attention mechanism. Please provide more explanations.
2.	The used five datasets are mainly based on single-modal input. In another related task of TV Show retrieval [1][2], the inputs are a video and its corresponding subtitle. Did the authors try the proposed method on this task and validate the its generalization ability?
3.	The authors are suggested to provide some failure cases and discuss the potential limitation of this work.
4.	Typos. In Line 11-12, amidsting -> amidst; in Line 27, two benchmark datasets -> five benchmark datasets.

[1] Tvr: A large-scale dataset for video-subtitle moment retrieval, ECCV 2020.

[2] I2Transformer: Intra-and Inter-relation Embedding Transformer for TV Show Captioning, IEEE TIP 2022.

**Suitability:**

3

---

### Meta-Review · Area_Chair_D85E · 2024-07-03

**Recommendation:** Accept (Poster)
**Confidence:** 5

**Metareview:**

This paper received 1 weak accept, 1 borderline accept, and 1 weak reject at the review period. Most of the reviewers acknowledged that the idea of the paper is interesting and the performance of the proposed method is good. After rebuttal, all the 3 reviewers kept their original rating in the final review, so 2 out of 3 reviewers are inclined towards accepting this paper, while there is still one reviewer who shows his(her) concerns in the number of parameters and model scales. The AC has carefully checked the comments, rebuttal, and feedback (if any) from the reviewers. Overall, this paper addresses an important problem of Text-Video Retrieval, of which the pros outweigh the cons. Therefore, the AC recommends accepting this paper for publication in ACM MM 2024 but highly encourages the authors to take into account the valuable comments from the reviewers, e.g., the writing.

---

### Meta-Review · Senior_Area_Chairs · 2024-07-10

**Recommendation:** Accept (Poster)
**Confidence:** 5

**Metareview:**

This paper received mixed ratings initially. After rebuttal, two reviewers tend to accept the paper while one still has some concerns. SAC and AC carefully checked the reviews and rebuttal and recommend acceptance of the paper.